# Angiotensin-II-Evoked Ca^2+^ Entry in Murine Cardiac Fibroblasts Does Not Depend on TRPC Channels

**DOI:** 10.3390/cells9020322

**Published:** 2020-01-29

**Authors:** Juan E. Camacho Londoño, André Marx, Axel E. Kraft, Alexander Schürger, Christin Richter, Alexander Dietrich, Peter Lipp, Lutz Birnbaumer, Marc Freichel

**Affiliations:** 1Pharmakologisches Institut, Ruprecht-Karls-Universität Heidelberg, INF 366, 69120 Heidelberg, Germany; 2DZHK (German Centre for Cardiovascular Research), partner site Heidelberg, 69120 Mannheim, Germany; 3Walther-Straub-Institut für Pharmakologie und Toxikologie, Ludwig-Maximilians-Universität, 80336 München, Germany; 4Medical Faculty, Centre for Molecular Signalling (PZMS), Institute for Molecular Cell Biology and Research Center for Molecular Imaging and Screening, Saarland University, 66421 Homburg/Saar, Germany; 5Laboratory of Neurobiology, NIEHS, North Carolina, USA and Institute of Biomedical Research (BIOMED), Catholic University of Argentina, Buenos Aires C1107AFF, Argentina

**Keywords:** TRPC channels, cardiac fibroblasts (CFs), Ca^2+^ release and Ca^2+^ entry, angiotensin II

## Abstract

TRPC proteins form cation conducting channels regulated by different stimuli and are regulators of the cellular calcium homeostasis. TRPC are expressed in cardiac cells including cardiac fibroblasts (CFs) and have been implicated in the development of pathological cardiac remodeling including fibrosis. Using Ca^2+^ imaging and several compound TRPC knockout mouse lines we analyzed the involvement of TRPC proteins for the angiotensin II (AngII)-induced changes in Ca^2+^ homeostasis in CFs isolated from adult mice. Using qPCR we detected transcripts of all *Trpc* genes in CFs; *Trpc1*, *Trpc3* and *Trpc4* being the most abundant ones. We show that the AngII-induced Ca^2+^ entry but also Ca^2+^ release from intracellular stores are critically dependent on the density of CFs in culture and are inversely correlated with the expression of the myofibroblast marker α-smooth muscle actin. Our Ca^2+^ measurements depict that the AngII- and thrombin-induced Ca^2+^ transients, and the AngII-induced Ca^2+^ entry and Ca^2+^ release are not affected in CFs isolated from mice lacking all seven TRPC proteins (TRPC-hepta KO) compared to control cells. However, pre-incubation with GSK7975A (10 µM), which sufficiently inhibits CRAC channels in other cells, abolished AngII-induced Ca^2+^ entry. Consequently, we conclude the dispensability of the TRPC channels for the acute neurohumoral Ca^2+^ signaling evoked by AngII in isolated CFs and suggest the contribution of members of the Orai channel family as molecular constituents responsible for this pathophysiologically important Ca^2+^ entry pathway.

## 1. Introduction

Remodeling processes comprising the activation of fibroblasts and the consequent changing in the extracellular matrix (ECM) composition are key processes during the development of different pathologies in several organs. Abnormal deposition of ECM proteins including collagens, or fibrosis occurs for example during non-alcoholic fatty liver disease (NAFLD) [1,2] or pathological cardiac remodeling presented by the failing heart or after myocardial infarction [3,4].

Among the different cell types found in the heart, cardiac fibroblasts (CFs) which represent the big majority are important mediators regulating the secretion of growth factors with direct impact on cardiomyocyte function, cardiac hypertrophy development and ECM formation during both normal cardiac growth and pathological conditions [5,6]. The term “cardiac fibroblast” groups a heterogeneous population of cells with probably multiple origins and different functional, morphological and molecular profiles that are observed depending on the status of the heart [7,8]. In the last years an eminent increased effort to characterize such cells has been evident; for example, to define differences between populations of resident fibroblasts, activated fibroblasts and myofibroblasts, the last ones being prominent in ECM deposition, secretion and tissue contraction [9,10,11]. Recently, matrifibrocytes were described, which develop from myofibroblasts and persist in the scar after myocardial infarction giving structural support [12]. Therefore, a multiple characterization of fibroblasts models such as primary cultured cells is a necessary step to understand the biology behind fibroblast functioning and required to utilize them in the development of therapeutic strategies, for example for heart disease management [13].

In CFs, different ion channels and ion transport pathways are found, and Ca^2+^ entry pathways are important in CFs and are considered to contribute to the development of fibrosis. In CFs, Ca^2+^ entry pathways include receptor-operated Ca^2+^ entry (ROCE) channels and store-operated Ca^2+^ entry (SOCE) channels [14,15]. ROCE channels are engaged via G-Protein coupled receptor-mediated activation of the Phospholipase C (PLC) (i.e., downstream of angiotensin II -AngII-) and receptor tyrosine kinase-mediated activation of the PLC signaling. Both result in hydrolysis of phosphatidylinositol-4,5-bisphosphate (PIP_2_) with formation of the second messenger diacylglycerol (DAG) and inositol 1,4,5- trisphosphate (IP_3_) [16,17,18]. SOCE is activated in response to depletion of the Ca^2+^ concentration within the lumen of the endoplasmatic reticulum (ER) [16,19]. This can be achieved by agonist evoked generation of inositol 1,4,5-trisphosphate (IP_3_) and subsequent ER Ca^2+^ release through the IP_3_ receptor (InsP_3_R) or experimentally by thapsigargin, a specific inhibitor of the sarco-endoplasmatic reticulum Ca^2+^ATPase (SERCA), leading to passive depletion of ER Ca^2+^ stores. SOCE is mediated by channels activated by emptying of intracellular Ca^2+^ stores and the best characterized of these SOC channels are the Ca^2+^ release-activated Ca^2+^ channels (CRAC channels) [16]. Orai proteins form highly Ca^2+^ selective SOC channels [20,21,22,23] and three isoforms (Orai1, Orai2 and Orai3) have been described [24,25]. However, the relative contribution of each of the three Orai proteins to endogenous SOC channels may differ between individual primary cells as revealed in different subsets of primary lymphocytes [26]. Ca^2+^ depletion is sensed by STIM proteins (STIM1 and STIM2) located in the sarcoplasmatic reticulum membrane [22,27]. After Ca^2+^ store depletion the conformation of STIM proteins changes, they oligomerize on the sarcoplasmatic reticulum’s surface and translocate to the plasma membrane at so called ER-PM junctions where STIM proteins are able to activate Orai proteins through direct contact [27,28,29,30]. AngII is a prominent effector regulating fibroblast cell functions including proliferation, migration and ECM deposition leading to development of fibrosis [3]. It has been shown that AngII treatment of rat CFs induces expression of the transforming growth factor β1 (TGF-β1) and engagement of the AngII-TGF-β1 axis stimulates collagen I expression and cardiomyocyte hypertrophy [31,32,33]. This concept was proven in an in vivo model with TGF-β1-deficient mice that do not develop cardiomyocyte hypertrophy after AngII stimulation [34]. The role of TGF-β1 in myocardial infarction and cardiac remodeling including fibrosis and its regulation through AngII signaling have also been a topic of reviews [35,36]. In addition to AngII receptors there are also other receptors reported in CFs that evoke G_q/11_-PLC signaling such as PAR receptors activated by molecules like thrombin. PAR1 is the most abundant PAR receptor in rat CFs and mediates pro-fibrotic responses like α-SMA expression [37]. Another PAR receptor, PAR 4, which is expressed in murine CFs is functionally up-regulated under high glucose conditions linked to diabetes [38]. PAR 4 is also activated by AngII and other inflammatory mediators [39]. Stimulation with thrombin elicits Ca^2+^ transients in cardiac chick fibroblasts [40] and it was reported that thrombin-evoked Ca^2+^ signals are mediated by TRPCs in endothelial cells [41].

All members of the TRPC (canonical) subgroup of the family of Transient Receptor Potential (TRP) channels can form receptor-operated cation channels mediating ROCE [17] or can be part of the store operated Ca^2+^ entry (SOCE). The TRPC subfamily comprises seven members in mammals (TRPC1 to 7). These channels exhibit six putative transmembrane domains with both C and N termini being intracellularly located and it is believed that TRP channels are assembled as homo- or hetero-tetramers to form cation selective channels [42,43,44]. Recently, it was demonstrated via quantitative high-resolution mass spectrometry analysis using subtype specific antibodies that TRPC1, TRPC4, and TRPC5 form heteromultimers with each other but not with other TRP family members in the mouse brain and hippocampus [45]. Corresponding analyses in other tissues including the heart were not reported yet, although numerous studies reported their expression and function in cardiomyocytes but also in non-myocytes including cultured fibroblasts [46,47,48,49,50,51,52].

Regarding the role of TRPCs in CFs it was shown that in freshly isolated rat CFs non-selective cation currents triggered by stimulation of the C-type natriuretic peptide (CNP) receptors show characteristics of TRPC-mediated ion currents [53]. Some publications establish TRPC3 as an important regulator of fibroblast function and fibrosis development based on observations assuming the blockade of TRPC3 by channel blockers like Pyr-compounds - mainly Pyr3- or based on the analysis of global TRPC3-KO mice [54,55,56]. The structurally most closely related channel to TRPC3 is TRPC6 which has been reported as an important regulator of myofibroblast differentiation in CFs, including TGF-β1-mediated up-regulation of α-SMA and AngII-induced collagen expression [57,58,59,60]. TRPC6 should also contribute to the increase of intracellular Ca^2+^ concentration elicited by OAG (1-Oleoyl-2-acetyl-sn-glycerol) in human ventricular fibroblasts [61]. In vivo inhibition of TRPC6 by BI-749327 reduces cardiac fibrosis produced by pressure overload as well as fibrosis developed in a model of renal injury [62]. TRPC1/C4-DKO mice show reduced cardiac fibrosis development after pressure-overload as well as reduced expression of collagen genes after chronic neurohumoral stimulation with AngII [63].

In this study we used Ca^2+^ imaging in primarily isolated CFs from several TRPC-deficient mouse lines to determine the involvement of TRPC proteins for the AngII-induced Ca^2+^ release and Ca^2+^ entry in murine isolated adult CFs. We were able to detect transcripts from all seven TRPC in isolated fibroblasts and we show that the AngII-induced Ca^2+^ signals are critically dependent on cell culture density and are inversely correlated with the expression of the myofibroblast marker α-smooth muscle actin. We found that AngII-induced Ca^2+^ transients and the AngII-induced Ca^2+^ release and Ca^2+^ entry are not affected in CFs isolated from mice lacking all seven TRPC proteins compared to control cells; but, this Ca^2+^ entry pathway was inhibited by the CRAC blocker GSK7975A. Finally, we conclude the dispensability of the TRPC channels for the acute neurohumoral Ca^2+^ signaling evoked by AngII or thrombin in isolated CFs leading an open question for the molecular identities responsible for this complex response essential for fibroblast function.

## 2. Materials and Methods

### 2.1. Animal Experiments

All animal procedures were approved and performed according to the regulations of the Regierungspräsidium Karlsruhe and the University of Heidelberg (T80/13, T87/14 and T85/15) and conform to the guidelines from Directive 2010/63/EU of the European Parliament on the protection of animals used for scientific purposes. The generation of TRPC1, TRPC2, TRPC3, TRPC4, TRPC5, TRPC6 and TRPC7 single knock-out (KO) mice was described previously [64,65,66,67,68,69,70]. TRPC compound KO mice, such as TRPC1/C4-Double KO (DKO), TRPC3/C6-DKO and TRPC-hepta KO (*Trpc1/2/3/4/5/6/7*^–/–^) mice were consecutively generated by crossing TRPC-single and TRPC-multiple KO mouse lines [71]. TRPC-KO mouse lines had a mixed 129SvEv/C57Bl6/N (TRPC-hepta KO) or 129SvJ/C57Bl6/N (TRPC1/C4-DKO and TRPC3/C6-DKO) genetic background; offspring from the F1 generation between 129SvJ and C57Bl6/N intercrosses were used as controls. Mice were maintained under specified pathogen-free conditions at the animal facility (IBF) of the Heidelberg Medical Faculty in a 12-h light-dark cycle, and water and standard food (Rod18, LASvendi GmbH, Germany) were available to consume ad libitum.

### 2.2. Isolation and Primary Culture of Cardiac Fibroblasts (CFs)

To isolate CFs, the hearts from mice between 6–8 weeks old were isolated and digested by retrograde perfusion in a Langendorff apparatus as previously described [63,72]. A modified procedure is briefly described below. Mice were sacrificed by cervical dislocation; the hearts were quickly exposed via thoracotomy and 1–1.5 mL ice-cold modified Tyrode’s solution (TS) containing (in mM): 134 NaCl; 4 KCl; 1.2 MgSO_4_; 1.2 Na_2_HPO_4_; 10 HEPES; 11 glucose; pH 7.35 adjusted with NaOH, osmolarity 280–300 mOsm/kg and supplemented with 1000 U heparin, were slowly injected into both ventricles. Afterwards the hearts were removed and transferred to ice-cold TS and cannulated (20G cannula) via the aorta, attached to a Langendorff apparatus, and retrogradely perfused with carbogen (5% CO_2_, 95% O_2_ maintained at RT ~24 °C)-saturated TS containing 1000 U/mL heparin, 200 µM EGTA and 10 mM 2,3-butanedione- monoxime for about 7 min (until the remaining blood was completely removed). The perfusion solution was then changed to constantly gassed (carbogen) and tempered (36 °C) TS containing a collagenase/ protease mixture (Liberase TM, Roche Applied Science, Pleasanton, CA, USA) at a final concentration of 133 µg/mL for 2–6 min (depending on size and age of the heart). The temperature of the perfusate was either maintained at 35 °C or RT (24–25 °C) but no obvious differences were observed on the isolated cells. Thereafter, the ventricles were isolated from the atria, dissected and transferred to TS supplemented with 5% bovine serum albumin (BSA), 12.5 µM Ca_2_Cl, and 1.4 ng/mL DNase I (Sigma-Aldrich Munich, Germany). The ventricles were cut into small pieces followed by gentle homogenization of the tissue using a cut and fire polished 1000 µL plastic tip. The suspension (final volume ~10 mL) was filtered through a nylon filter (pore size 150 µm, Sysmex, Norderstedt, Germany) and maintained for 10 min (37 °C) in the same solution for sedimentation; cardiomyocytes were mainly concentrated in the pellet. Fibroblasts remaining in the supernatant were transferred to a 15 mL tube and remaining cardiomyocytes were removed by a short centrifugation step of 48× *g* (1 min, Megafuge 1.0 R, Heraeus, Hanau, Germany). The supernatant was transferred into a new tube and the cells were concentrated in a pellet by centrifugation (324× *g*, 10 min); after removal of the remaining TS the cells were washed once with medium 199 (FG0615, Merck, Darmstadt, Germany). Finally, cells were gently resuspended in 5 mL of medium 199 supplemented with 2.5% penicillin/streptomycin (151401222, Gibco, ThermoFisher Scientific, Waltham, MA), 50 µg/mL kanamycin (A1493, Applichem, Darmstadt, Germany), 0.1 µM insulin (I6634, Sigma-Aldrich) and 10% FCS (10270-106, Gibco). Cells were seeded on a 25-cm^2^ culture flask (one heart per flask) and placed in a cell incubator (5% CO_2_, 37 °C) for 2 h. Thereafter, the medium was changed to remove remaining tissue and not attached cells. After a period of 24 h of incubation the cells were de-attached from the culture flask by trypsin (1% in DPBS and 0.2 mM EDTA) treatment (~3 min). After washing the cells with medium 199 and determining the cell number, the harvested cells were resuspended in a desired concentration. To obtain cells on high cell density conditions (>300 cells/mm^2^) the CFs were resuspended at a concentration of 1000 cells/µL, then a drop of 10 µL from the cell suspension (small volume to ensure high density) was placed on a glass coverslips, the cells were let to attach for 1–2 h in the incubator; after the attachment period the medium was replenish (1–2 mL) and the cells were further cultured until its analysis. For low density cultures CFs were resuspended in 1.2 mL medium and the cells were seeded on glass coverslips (100–200 µL/well) where a silicone isolator (Grace Bio-Labs, Sigma-Aldrich) with a free area of 100 mm^2^ was placed. After 2 h the medium was replenished and cells were cultured until analysis. For expression analysis via qPCR the cells were cultured in high density conditions (see above) on coverslips pre-coated with Extra Cellular Matrix (ECM Gel from Engelbreth-Holm-Swarm murine sarcoma, E1270, Sigma-Aldrich). After 6 days, CFs were washed with DPBS followed by detachment of cells with trypsin. After centrifugation (300× *g*, 10 min), the cell pellet was resuspended in DPBS, containing 0.1% BSA, 25 mM HEPES and 1 mM EDTA. Finally, the supernatant was removed after centrifugation and the cell pellets were snap frozen in liquid nitrogen and stored at −80 °C until RNA isolation.

### 2.3. RNA Isolation and qPCR Analysis

RNA isolation from Langendorff heart-derived fibroblasts was performed using the Direct-Zol RNA MicroPrep (Zymo Research, Freiburg im Breisgau, Germany) including DNA digestion and following the instructions of the manufacturer. Integrity and quantity of the RNA samples were determined by microfluidic analysis (Bioanalyzer 2100, Agilent Technologies, Santa Clara, CA, USA) and fluorometry (qbit assay, ThermoFisher Scientific), respectively. cDNA synthesis from 700–1000 ng of RNA was carried out using a SuperScript first-strand synthesis system for RT-PCR (SensiFAST BIO-65054, BIOLINE, London, UK). 30 ng cDNA from each sample were used as template. In brief, qPCR was performed with a Universal Probe Library (Roche) (probes and primers are listed in Appendix A) by using Roche FastStart Essential DNA Probes MasterMix (Roche 06402682001) and detection on a Light Cycler96 (Roche). Thermal cycling conditions include: Initial denaturing at 95 °C for 600 s, followed by 40 cycles of amplification (each cycle: 10 s/95 °C, 30 s/60 °C). All PCR reactions were performed in duplicate and normalized to the expression of the house keeping genes H3F3A, AIP and CXXC1.The standard curve quantization method was used for gene expression measurements. Values are expressed as relative expression and calculated as follows: Relative Expression= (Mean E^-Cq^/Mean E_HKG_^-Cq^)/((Mean E_HKG_/100+1)^-Cq^ × 10^4^), being E the efficiency, Cq the cycle of quantification and HKG refers to housekeeping genes. For visualization of qPCR products 10 µL from the PCR reactions were run on 2% agarose gels at 100 V during 20 min; images were obtained and digitalized with a GelDoc™ XR system (Bio-Rad, Feldkirchen, Germany).

### 2.4. Immune Fluorescence Staining

Primary CFs isolated from WT mice (either C57BL6/N or the F1) were cultured on glass coverslips. Before cell fixation the samples were rinsed in cold PBS (in mM, 137 NaCl, 2.7 KCL, 8.1 Na_2_HPO_4_, 1.8 KH_2_PO_4_, pH 7.4, sterile filtered). Cell fixation was done by incubation in 4% Paraformaldehyde prepared in PBS over 10 min at 4 °C and subsequently the cells were washed 2 × 5 min by submerging them in cold PBS. The samples were transferred into pre-chilled acetone (−20 °C) for exactly 5 min (step omitted when CD31 was used) and washed for 5 min 3 × with cold PBS. To block unspecific binding, the samples were incubated for 1 h (RT) in 1% BSA prepared in PBST (0. 1% *v/v* Tween20 in PBS or only PBS for CD31) including 0.3 M glycine which reduces the background by binding to free aldehyde groups. Between 50–100 µL from the primary antibody solved in 1% BSA (PBST or only PBS for CD31) were added and the cells were incubated at room temperature and protected from light in a humid chamber. The following primary antibodies were used: Anti-P4HB (11245-AP, Acris, Herford, Germany), anti-DDR2 (sc-7555, Santa-Cruz, Dallas, TX, USA), anti-CD31 clone P2B1 (ab24590, abcam, Cambridge, UK) used as endothelial marker, anti-smooth muscle α2-actin (ab15734, abcam), anti-αSMA clone 1A4 (A2547, Sigma-Aldrich) and anti-α-actinin clone EA-53 (A7811, Sigma-Aldrich). After incubation with the primary antibody, three washing steps of 5 min each with cold PBS were done followed by the incubation with the secondary antibodies (Appendix A) at room temperature and protected from light. The secondary antibody mixtures were decanted and three 5 min-washing steps with cold PBS were performed. To stain the nuclei the cells were incubated for 5 min with DAPI 1.5 µg/mL in PBS. Finally, the coverslip were mounted on glass slides using an anti-fade mounting medium (Vectashield, Linaris, Dossenheim, Germany or self-made solution: 6 g glycerin, 2.4 g Mowiol 4-88, 6 mL ddH_2_O, 12 mL Tris-HCl 0.2 M pH 8.5 and DABCO 25 mg/mL) and were stored at 4 °C protected from light until analysis. Concentration and incubation times for each antibody used are depicted in Appendix A. As positive control for the selected markers freshly isolated mouse cardiomyocytes, freshly isolated ileum smooth muscle cells (iSMC) and mouse aortic endothelial cells (MAEC) were prepared as previously described [55,61,62]. Negative controls omitting the primary antibody were included and processed the same. For the fluorescence analysis two different setups were used. First, an AxioVert 200 M inverted microscope (Zeiss, Jena, Germany) equipped with a HXP120 fluorescence lamp (Kübler codix, Leistungselektronik JENA GmbH, Jena, Germany), a digital camera AxioCam MRm (Zeiss), filters (AHF analysentechnik AG, Tübingen, Germany) for FURA (DAPI), GFP (Alexa Fluor-488) and Alexa-594 was used. Alternatively an Axio Observer Z.1 microscope equipped with DG-4 light source (Sutter Instruments, Novato, CA, USA), an AxioCam MRM camera (Zeiss) and, HC Basic (F26-510, DAPI), HC Basic TxRed (F26-518) and HC EGFP (F36-525) filter sets was used. Images were digitalized using the AxioVision v4.7.2 software (Zeiss).

### 2.5. Calcium Imaging

One day prior (at least 24 h before) to calcium measurements cells were changed to a medium without FCS that was replaced by to 0.01% BSA (A7906, Sigma-Aldrich). Cells were incubated with 5 μM fura-2 acetoxymethyl ester (dissolved in 20% *w*/*v* Pluronic, F-127 Sigma-Aldrich in DMSO) for 30 min at room temperature in a physiological solution that contained in mM: 134 NaCl, 4 KCl, 1.2 MgSO_4_, 1.2 Na_2_HPO_4_, 2 CaCl_2_, 11 glucose and 10 HEPES. After incubation the cells were rinsed 3 times with physiological solution and the glass coverslips were transferred into a measuring chamber (AttoFluor^®^, MolecularProbes, ThermoFisher Scientific, Waltham, MA). Changes in intracellular Ca^2+^ concentration were measured using an inverted microscope (Axio Observer-A1 or Axio Observer Z1, Zeiss) equipped with either a Monochromator (Polychrome V, Till Photonics, Planegg, Germany) or a monochromator-based imaging system consisting of a Lambda DG-4 Plus Light source (Sutter Instrument), a Filterset Fura 2 HC (Nr.: H76-521, AHF, Tübingen, Germany), a 20× objective (NA 0.75, Fluar, Zeiss) and a CCD camera (Axiocam MMR, Zeiss). Image acquisition and digitalization were done with the Physiology Module from the Axiovision 4.8.2 software (Zeiss). Fluorescence at 510 nm was measured during alternative excitation at 340 nm and 380 nm. The excitation and emission detection cycles were done every 5 s. After correction for the background fluorescence signals at both excitation wavelengths, the fluorescence ratio (F340/F380) was calculated and analyzed using OriginPro 2015G (OriginLab, Northampton, MA, USA). For Ca^2+^ measurements without extracellular Ca^2+^, CaCl_2_ was omitted from the physiological solution and was it replaced by EGTA as depicted on the figures. Perfusion and changing of the solutions was done by a gravimetric perfusion system coupled to a suction vacuum pump.

### 2.6. Agonist and Blockers

The following compounds were used: angiotensin II (Cat. Nr. A9525, Sigma-Aldrich), thrombin (Cat. Nr. T4648, Sigma-Aldrich), TGF-^®^1 (Cat. Nr. 101-B1-001, R&D Systems/Biotechne, Wiesbaden, Germany), GSK7975A (2,6-difluoro-N-(1-(4-hydroxy-2-(trifluoromethyl)benzyl)-1*H*-pyrazol-3-yl)benzamide; Cat. Nr. AOB4124, AOBIOUS Inc., Gloucester, MA, USA), SAR7334 (4-(((1*R*,2*R*)-2-((*R*)-3-aminopiperidin-1-yl)-2,3-dihydro-1*H*-inden-1-yl)oxy)-3-chlorobenzonitrile, provided by Bayer Pharma AG, Leverkusen, Germany), T320722 (5-chloro-2-(4-methylpyridin-1-yl)-*1H*-benzo [D]imidazole, Cat.Nr. T320722, Sigma-Aldrich). In experiments where the TRPC- or CRAC-antagonists were used CFs were pre-incubated for 10 min in TS solution containing the blockers which were also included in the solutions used during the Ca^2+^ imaging experiments.

### 2.7. Statistical Analysis

Data were analyzed and processed with Excel 2007 (Microsoft, Redmond, WA, USA) and OriginPro 2015G ver. 9.2.257 (OriginLab). Values are shown as mean ± standard error of the mean (SEM) except for Figure 1C where standard deviation (SD) is shown. Statistical significances were assumed when *p* < 0.05. *p*-values are depicted as * *p* < 0.05; ** *p* < 0.01, *** *p* < 0.001. Statistical analysis was performed using the two-tailed unpaired Student’s *t*-test except in Figure 1C where a Kruskal-Wallis test followed by Dunn’s test for multiple comparisons were performed due to differences in the variance among the groups.

## 3. Results

### 3.1. Characterization of Primary Isolated Murine Cardiac Fibroblasts (CFs)

To determine the role of TRPC channels for the angiotensin II-induced Ca^2+^ response in isolated primary murine CFs we first determined the conditions of cell culture and analysis using CFs obtained from adult wild type (WT) mice. For that purpose cells isolated by a Langendorff-based perfusion system and subjected to a pre-plating strategy were cultured in different cell densities and subsequently analyzed with respect to their Ca^2+^ homeostasis (Figure 1A).

As initial step in our analysis the cells were characterized by immunocytochemistry (Figure 1B). As markers for fibroblasts we used antibodies against the prolyl 4-hydroxylase (P4HB) and the discoidin domain-containing receptor 2 (DDR2), which are considered as specific markers for these cells [73,74]. In addition, we used anti-α-actinin as marker for cardiomyocytes, CD31 for endothelial cells and α-smooth muscle actin for smooth muscle cells or for myofibroblasts, and included a corresponding positive control for each of these markers: cardiomyocytes, mouse aortic endothelial cells and ileum smooth muscle cells. We observed that our cultured primary fibroblasts were positive for both P4HB and DDR2 but negative for CD31 and α-actinin as expected (Figure 1B). Regarding α-smooth muscle actin the CFs were negative for this marker only when cultured in a density higher than 300 cells/mm^2^. When cultured on lower density conditions the cell staining was positive for α-smooth muscle actin (Figure 1B, Figure 2C,D and Appendix A). This means that CFs easily differentiate into a myofibroblast-like phenotype when they are seeded at low density. Using cultured fibroblasts we also quantified expression of the transcripts encoding the seven members of the TRPC ion channel family. We could detect expression of all seven TRPC members being *Trpc1*, *Trpc3* and *Trpc4* most prominently expressed and TRPC4 being significantly higher expressed compared to *Trpc*5 and *Trpc*6 in CF cultures at day 7 after isolation (Figure 1C).

### 3.2. Differential Changes in Intracellular Ca^2+^ Concentration in CFs Depending on the Density of the Culture

To monitor changes in the intracellular Ca^2+^ concentration in CFs we used Fura-2 loaded cells. We performed two kinds of protocols: With the first protocol agonist-induced Ca^2+^ transients in the presence of extracellular Ca^2+^ [2 mM] were investigated (Figure 2A,B). With the second protocol the agonist-induced Ca^2+^ release from internal stores in the absence of extracellular Ca^2+^ and the subsequent agonist-induced Ca^2+^ entry in the presence of extracellular Ca^2+^ were analyzed (Figure 2C,D). We observed that the AngII-induced Ca^2+^ transients were significantly larger when the cells were maintained on high density conditions (Figure 2A,B); in addition, the AngII-induced Ca^2+^ release from intracellular stores was also larger in cells kept in high density conditions and AngII-induced Ca^2+^ entry was only measureable in CFs cultured in these conditions (Figure 2C,D). These observations correlate inversely with the expression of the α-smooth muscle actin (Figure 2C,D and Appendix A). Taken together, our observations lead to the conclusion that the cell density is an important component that defines the cell morphology, protein expression and agonist-induced Ca^2+^ signaling in primary cultures of CFs. Using the concept that the AngII-induced Ca^2+^ transients are more prominent under high density, we tested if TRPC1/C4 deletion could alter these responses; however, no differences were observed between WT and DKO-CFs after 4, 5 or 6 days in culture (Appendix A). Additionally, we tested the Ca^2+^ transients evoked by different Thrombin concentrations (Appendix A) and found also no differences between fibroblasts from WT and TRPC1/C4-DKO mice (Appendix A). These observations indicate the dispensability of TRPC1/C4 in the acute Ca^2+^ signaling in isolated CFs.

### 3.3. Effect of TRPC3/C6 Genetic Deletion or Its Pharmacological Inhibition with SAR7334 on the AngII-Induced Ca^2+^ Release and Ca^2+^ Entry

After defining that AngII-induced Ca^2+^ entry with a robust Ca^2+^ release is only observed in fibroblasts kept at high density and TRPC3 and TRPC6 being described as important regulators of fibroblast function [15,47,48,51], we studied the effect of genetic deletion of TRPC3/C6 proteins or its acute inhibition with the described TRPC3/C6/C7 blocker SAR7334 [75] on AngII-induced Ca^2+^ release and Ca^2+^ entry. The genetic deletion of both TRPC3 and TRPC6 did not affect Ca^2+^ release or Ca^2+^ entry induced by AngII in primary CFs. No difference between WT and TRPC3/C6-DKO CFs on the AngII-induced Ca^2+^ signaling were observed after 5 (Figure 3A) or 6 days (Appendix A) in culture. Moreover, the acute blockage of TRPC3/C6/C7 channels by SAR7334 had no effect on the AngII-induced Ca^2+^ release and Ca^2+^ entry in WT cells or in cells isolated from TRPC3/C6-DKO mice (Figure 3B and Appendix A). These results indicate that there is no involvement of TRPC3/C6 proteins in the acute AngII-induced Ca^2+^ release and Ca^2+^ entry in primary fibroblasts and moreover that SAR7334 does not have unspecific effects on AngII-induced Ca^2+^ release and Ca^2+^ entry in this cell system as no difference was observed between WT and TRPC3/C6-DKO CFs treated with the blocker.

Because it was recently shown that expression of *Trpc6* is upregulated by TGF-β treatment and that TRPC6 channels play a role in TGF-β induced myofibroblast differentiation [58] we pre-incubated CFs with TGF-β to determine the effect of TGF-β treatment on Ca^2+^ signaling. After TGF-β pre-treatment we observed an increased expression of α-smooth muscle actin even though the cells were kept on high density conditions during 5 or 6 days in culture (Figure 3C and Appendix A). In line with our findings in Figure 2, we observed in WT cells that the increased α-smooth muscle actin expression was inversely correlated with the magnitude of the AngII-induced Ca^2+^ release and Ca^2+^ entry in adult CFs (Figure 3C). This points out that TGF-β leads to differentiation of our cultured fibroblasts to a more myofibroblast-like phenotype and significantly reduces the AngII-induced Ca^2+^ signaling like it was observed under low-density conditions.

### 3.4. Dispensability of All TRPC Proteins for the Acute Ca^2+^ Release and Ca^2+^ Entry Induced by AngII in Cardiac Fibroblasts

We could show that members of two subgroups of the TRPC family, TRPC1/C4 and TRPC3/C6, were not involved in the acute AngII-induced changes in intracellular Ca^2+^ concentration in primary fibroblasts.

To continue with our study, we additionally tested if any other member of the TRPC subgroup such as e.g., TRPC4/C5 could be part of the AngII-induced Ca^2+^ signaling in CFs. For this purpose, we took advantage of the recently described TRPC blocker T320722 [76] with relative specificity for TRPC4 (IC_50_ 4.1 µM) and TRPC5 (IC_50_ 3.1 µM) compared to TRPC3 (IC_50_ 30.4 µM) and TRPC6 (IC_50_ 57.1 µM). Pre-treatment (10 min) of CFs with 10 µM T320722 had no effect on the AngII-induced Ca^2+^ release but it was able to significantly reduce the AngII-induced Ca^2+^ entry in WT cells (Appendix A). To further test this observations and because the deletion of one TRPC protein might be compensated by the upregulation of other TRPCs [69] we decided to analyze the Ca^2+^ signaling in fibroblasts in which none of the TRPC channel proteins is expressed using cells isolated from TRPC-hepta KO (*Trpc1/2/3/4/5/6/7*^–/–^) mice [71]. The acute Ca^2+^ transients evoked either by AngII or by thrombin were not different between WT and TRPC-hepta-KO CFs after 5 or 6 days in culture (Figure 4A,B and Appendix A). Moreover, in the absence of all TRPC proteins neither the acute AngII-evoked Ca^2+^ release nor the AngII-evoked Ca^2+^ entry were impaired in comparison with WT control cells (Figure 4C and Appendix A). Together these results indicate that deletion of all TRPC-proteins in CFs does not impair the acute AngII-induced Ca^2+^ release and entry in isolated CFs. To further explore other possible molecular entities responsible for the acute AngII-induced Ca^2+^ entry in CFs, we performed experiments in CFs from WT mice in the presence of the CRAC inhibitor GSK7975A [77] that was characterized in pancreatic acinar cells and mast cells [78,79]. Pre-incubation with the CRAC blocker was able to completely abolish the Ca^2+^ entry initiated by 100 nM AngII stimulation (Figure 4D), indicating the involvement of proteins from the Orai family in this Ca^2+^ entry pathway.

## 4. Discussion

There are several reports analyzing the role of TRPC proteins in cardiac fibroblasts (CFs) and in fibrosis development using TRPC antagonist, siRNA or shRNA; however, the causal contribution of TRPC isoforms for Ca^2+^ entry following stimulation with neurohumoral agonists relevant during development of cardiac remodeling in primary cardiac fibroblast isolated from TRPC-KO mice has not been studied so far. In this study we therefore aimed to assess the causal contribution of TRPC proteins for Ca^2+^ entry following acute AngII application by comparative analysis of primary isolated CFs from several *Trpc*-deficient mouse models and WT controls. We showed that CFs isolated from global TRPC-hepta KO (*Trpc1/2/3/4/5/6/7*^–/–^) mice exhibit no impairment in AngII-induced Ca^2+^ entry nor Ca^2+^ release.

To define the CF cell model used in this study we could prove a high homogeneity of our cultured cell population by positive immunostaining for the established CF-specific markers P4HB and DDR2 that was comparable with findings from other studies using similar approaches [74,80]. We avoided the use of the frequently used marker vimentin because it is also expressed in endothelial cells as well as in macrophages [73,80]. Comparison of two different culture conditions that we defined as low and high density, showed that a significant AngII-evoked Ca^2+^ entry was only observed at the increased cell density >300 cells/mm^2^ and that the Ca^2+^ release from intracellular stores was largely increased under high density condition. When the maximal peaks of the Ca^2+^ signals of the Ca^2+^ release and the Ca^2+^ entry are compared, the maximum peak of the Ca^2+^ entry is much lower than the one from the Ca^2+^ release. However, the Ca^2+^ release signals decay much faster in comparison to the Ca^2+^ entry; the Ca^2+^ entry is maintained elevated at least over minutes and it could account for higher Ca^2+^ concentrations over time triggering specific transcriptional programs and/or extracellular matrix production. Obviously, one or more steps in the signaling cascade triggered by AngII in CFs are enhanced under these high-density conditions. In addition, the magnitude of the Ca^2+^ signals induced by AngII is inversely correlated with the expression of the myofibroblast marker α-SMA. These correlation between Ca^2+^ signaling and cell density is not surprising due to published observations e.g., from endothelial cells where the importance of cell to cell contact for Ca^2+^ signaling was shown [81]. Defining and characterizing the cell culture conditions is an essential prerequisite to obtain reliable and reproducible experimental results in primary cells, and not only for the analysis of Ca^2+^ signaling. For example, changes in the starvation conditions, which were also optimized in our approach in CFs, have profound effects on the secretome of human prostatic stromal myofibroblasts and other cell types [82]. Despite that we did not look to the impact of the culture substrate on the Ca^2+^ signaling, this is another important variable determining the responses in primary cells; different culture substrates have significant effects on expression profiles as revealed by a transcriptome analysis from human dermal fibroblasts cells [83].

With our established protocol to culture and measure the AngII-induced Ca^2+^ release and Ca^2+^ entry, we first examined the expression of the TRPC channels and found transcripts from all seven *Trpc* genes with *Trpc1*, *Trpc3* and *Trpc4* being most abundant. Since it has been reported that TRPC3 and TRPC6 are involved in fibroblasts function we first analyzed CFs from TRPC3/C6-DKO mice, to also circumvent possible compensation of TRPC6 in TRPC3-KOs as described [69]. We observed no alterations in AngII-induced Ca^2+^ entry or Ca^2+^ release compared to WT CFs. In addition, the TRPC3/C6/C7 blocker SAR7334 [75] elicited no changes in the Ca^2+^ responses. Based on our results obtained with this TRPC3/C6/C7 channel inhibitor it can be assumed that also TRPC7 has no essential role in the AngII-induced Ca^2+^ signals, because the acute blockage of all three channels (in WT cells) or the genetic inhibition of TRPC3/C6 plus pharmacological inhibition of TRPC7 (TRPC3/C6-DKO cells) did not affect the Ca^2+^ responses. In contrast to our results, it is published that targeting TRPC3 channels by pharmacological inhibition using Pyr3 in cardiac fibroblast from adult rats is able to blunt AngII-induced Ca^2+^ entry and α-SMA expression [54]. Moreover, in in vivo fibrosis models it was shown that Pyr3 treatment reduces the increase in proliferation and α-SMA expression observed in left atrial fibroblasts from an atrial fibrillation model in dogs [54], the cardiac fibrosis and expression of fibrosis associated genes after pressure overload, and the TGF-β-induced CTGF and αSMA expression in human fibroblasts [55]. In line with these observations another Pyr compound, Pyr10, prevents myocardial fibrosis in a hypertension model [56]. Discrepancies between our study and published data can be accounted to the fibroblast’s origin (mouse, human or rat) or to the culture conditions. Unfortunately, in the in vivo studies published with TRPC3-KO mice no experiments with isolated CFs and no Ca^2+^ imaging experiments were reported. Remarkably, both Pyr compounds also block Orai1-mediated store-operated Ca^2+^ currents, with Pyr10 being more selective for TRPC3 (Pyr3: EC_50_ ROCE-TRPC3 0.54 µM and EC_50_ SOCE 0.54 µM; Pyr10 EC_50_ ROCE-TRPC3 0.72 µM and EC_50_ SOCE 13.08 µM) [84]. Furthermore, inhibition by siRNA of Orai1 or STIM1 in rat CFs reduces the AngII-induced Ca^2+^ entry, as well as the AngII-induced increase in expression of α-SMA, CTGF and Fibronectin [85]. These results are in accordance with our observations with the CRAC blocker GSK7975A that completely inhibited the AngII-induced Ca^2+^ entry. TRPC6 is regarded as an important regulator of fibroblasts function especially in terms of myofibroblast differentiation [57,58]. Approaches down-regulating TRPC6 expression showed reduction in the TGF-β1-mediated up-regulation of α-SMA in human CFs from the right ventricle [59], reduction in the AngII-induced increase in collagen expression in rat CFs [60] and reduction in the OAG (DAG analogue)-induced elevation of Ca^2+^concentration in human ventricular fibroblasts [61]. One can assume that AngII-evoked activation of Gq-coupled receptors and downstream signaling implicates DAG formation which can directly activate TRPC6-containing channels [86]; however, in TRPC3/C6-DKO the AngII-induced Ca^2+^ signaling was not impaired, but ROCE induced by AngII following siRNA-mediated knock down of TRPC6 was not analyzed in the study from Ikeda and co-workers [61]. In contrast to our observations Ikeda and co-workers also showed that treatment with TGF-β enhanced the AngII-induced Ca^2+^ entry, but using a 100-fold higher AngII concentration in human cells. In vivo evidence for the role of TRPC6 during fibrosis development comes from observations using TRPC6-KO mice. TRPC6-deficient dermal fibroblasts isolated from TRPC6-KO mice are refractory to TGF-β and AngII-induced transdifferentiation (58); similar to that study we also observed that TGF-β treatment leads to a differentiation of the CFs to myofibroblasts with increased expression of α-SMA [58]. TRPC6-KO mice show impaired dermal and cardiac wound healing [58]. In that study the Ca^2+^ entry in CFs produced by inhibiting the Sarcoplasmic/endoplasmic reticulum calcium ATPase (SERCA) was enhanced by TGF-β or AngII treatment and it was attributed to increased TRPC6 expression. In our experiments the Ca^2+^ entry after TGF-β pre-treatment was reduced in WT CFs and it was not different in the absence or pharmacological inhibition of TRPC6. Moreover, in primary murine lung fibroblasts AngII treatment does not affect TRPC6 levels whereas TGF-β changes TRPC6 expression levels [87]. Those differences can most likely be attributed to cell origin, cell culture conditions and/or different Ca^2+^ entry mechanism between the entry evoked by AngII or by SERCA blockage. Inhibition of TRPC6 by BI-749327 was shown to reduce cardiac fibrosis produced by pressure overload as well as fibrosis developed in a model of renal injury [62]; however, this blocker has not been validated in TRPC6-deficient cells or regarding its specificity on TRPC vs. its action on channels composed of Orai proteins. Beyond TRPC3 and TRPC6 we also analyzed with our experiments in TRPC-hepta KO and TRPC1/C4-DKO CFs the contribution of TRPC1 proteins to the AngII-induced Ca^2+^ signaling. Other groups also attempted to determine the activity of TRPC1 in right atrial human fibroblasts; however, the treatment with siRNAs against TRPC1 did not alter Ca^2+^ influx that was evoked in these fibroblasts by application of a bath solution containing 20 mM Ca^2+^ [88]. 

The fact that we observed no differences in acute AngII-evoked Ca^2+^ signaling in the absence of both TRPC3/C6, and even in the absence of all seven TRPC proteins in CFs does not completely preclude that these TRPC proteins might be involved in the regulation of fibrosis-related gene expression in CFs. It is possible that conformational changes and signaling mediated thereby independent from ionic signaling conveyed by a channel protein can also play a role in stimulating transcription [89]. Along these lines it was recently shown that TRPC6 proteins are signaling via interaction with calpain independent of their channel function [90]. 

Other candidates outside the TRPC group that could mediate the AngII-induced ROCE are found in other TRP subfamilies like TRPV, TRPM or TRPA, therefore below we discuss their implications in CFs functioning and fibrosis. For several TRP proteins different mechanisms for their involvement in fibroblast function and fibrosis development have been proposed. In CFs from TRPV1 over-expressing mice the sustained Ca^2+^ rise provoked by capsaicin is enhanced; additionally, in those mice the Isoproterenol-induced cardiac fibrosis and cardiac hypertrophy in vivo are reduced [91]. In vitro, the AngII-induced proliferation of CFs is reduced by capsaicin in cells from WT but not from TRPV1-deficient mice [92]. In rats, the TRPV3 activator Carvacrol is capable to increase the fibrosis produced by aortic banding and it potentiates the AngII-induced increase in Ca^2+^ concentration as well as the in vitro proliferation of CFs. In contrast, treatment with siRNA directed against TRPV3 reduces the AngII-induced collagen expression [93]. Another TRPV channel, TRPV4, which determines myofibroblast differentiation and fibrosis in the lung [94], was also shown to mediate TGF-β1-induced differentiation of CFs into myofibroblasts [95]. The TRPV4-specific antagonist AB159908 as well as siRNA knockdown of TRPV4 significantly inhibit TGF-β1-induced differentiation as measured by incorporation of α-SMA into stress fibers of TGF-β1 treated fibroblasts. TGF-β1 treated fibroblasts exhibit enhanced TRPV4 expression and TRPV4-mediated calcium influx in CFs compared to untreated controls [95]. Stimulation of murine CFs isolated from WT mice with the TRPV4 agonist GSK-1016790A results in a significant Ca^2+^ rise, but this response is entirely absent in CFs from TRPV4-KO mice indicating that TRPV4 is functionally active in these CFs [96] and could be another candidate for the AngII-induced Ca^2+^ signaling. However, the role of TRPV4 for Ca^2+^ entry in CFs triggered by neurohumorally acting agonist has not been reported yet.

A hypoxia-evoked cation current was described in cultured primary fibroblasts from adult rat hearts and siRNA against TRPM2 prevented the development of such current [97]. H_2_O_2_ induces a higher Ca^2+^ elevation in hypoxia-exposed adult rat CFs compared to normoxia-exposed cells, which was also attributed to TRPM2 [97]. In addition, TRPM7-like currents have been characterized in CFs from different species including mouse, human and rat [88,98,99,100]. Using also hypoxia as stimulus, it was proposed that the increased TRPM7 expression and activity in rat CFs during hypoxia is blocked by astragaloside-IV (most active component of Chinese sp. *Astragalus*), and that its inhibition by siRNA prevents fibrotic features in NIH-3T3 mouse fibroblasts [101]. Additionally, in rat CFs H_2_O_2_ and AngII also are able to provoke a sustained increase in the intracellular Ca^2+^ and elevated expression of fibrogenic growth factors (e.g., α-SMA, TGF-β1) that were either blunted or reduced in fibroblasts in which TRPM7 was silenced via shRNA [102]. In human right atrial fibroblasts similar results were observed. Knockdown of *Trpm7* mRNA with shRNA treatment reduces TGF-β1-induced differentiation and proliferation [88]. Using neonatal rat CFs and siRNA-mediated knockdown of TRPM7 it was recently proposed that TRPM7 regulates fibrotic features through the microRNA-135a [100]. Using macrophages from mice with deletion of the TRPM7-kinase domain it was revealed that such cells are able to induce a fibrotic phenotype (i.e., increased TGF-β) in CFs from WT mice and interestingly this effect was prevented by MgCl_2_ treatment [103]. These results lead to the conclusion that TRPM7 channels in other cells than CFs might play a role in fibrosis development. TRPA1, the only member of the TRPA group, has also been studied in fibroblasts. In primary human ventricular CFs methylglyoxal provokes a sustained increase in the intracellular Ca^2+^ concentration that is largely reduced by treatment with HC030031, a selective TRPA1 antagonist or by TRPA1 knockdown using siRNA [104]. In CFs from TRPA1-deficient mice the transdifferentiation evoked by TGF-β is strongly reduced; in vivo*,* cardiac fibrosis is reduced in TRPA1-KO mice after myocardial infarction but is increased in WT mice treated with the TRPA1 agonist cinnamaldehyde [105]. However, it remains still unclear whether development of cardiac fibrosis in these experiments is mediated by TRPA1 channels present in CFs or other cell types.

A main general limitation in studies about the role of TRPC channels for complex in vivo phenotypes and associated cellular mechanisms is the use of global KO mice. Global deletion of TRPC proteins during early development might evoke mechanisms and pathways that compensate for the loss of TRPC-containing channels. Furthermore, differences in processes depending on cell-cell interactions including Ca^2+^ signaling might occur. For example, TRPC1/C4-DKO mice show reduced development of cardiac fibrosis after pressure-overload as well as reduced expression of collagen genes after chronic neurohumoral stimulation with AngII most likely due to a reduced background Ca^2+^ entry in cardiomyocytes [63]. Nevertheless, this phenotype might also rely on TRPC1/C4 activity in other cells, particular cardiac non-myocytes, or depend on an in vivo cross-talk between cardiomyocytes and non-myocyte cells in the heart. However, the present study reduces the likelihood that AngII-triggered Ca^2+^ dependent processes in CFs, such as expression of fibrogenic growth factors and development of tissue fibrosis, are mediated by channels composed of TRPC proteins. To circumvent such problems and to address such questions the use of new conditional *Trpc* alleles (i.e., TRPC1/C4) that allow for a cell type-specific deletion of selective TRPC proteins would be a promising approach. Nonetheless, such approaches can be challenging in CFs due to the diversity of CFs found in the healthy heart and during disease progression [11]. Alternatively the acute inhibition of TRPC channels by inhibitors might be used as an experimental setup to study Ca^2+^ homeostasis in isolated cells such as CFs. Here we observed that the TRPC blocker (T320722) reduces the AngII-induced Ca^2+^ entry; however, our interpretation in light of the observation from TRPC-deficient cardiac fibroblasts is that such effect is more likely due to unspecific effects of that compound rather than the inhibition of a TRPC4/C5-mediated Ca^2+^ entry. Recently, an inhibitor of TRPC4- and TRPC5-homomeric channels as well TRPC1/TRPC4- and TRPC1/TRPC5-heteromeric channels has been developed [106,107]. This blocker (compound C31 or pico145) is highly potent (IC_50_ in the picomolar range) and specific within the TRP channel family. However, its effectiveness in primary cells and in vivo has not yet been demonstrated. Apart from C31, other TRPC channel blockers such as ML204 and recent new derivates of C31 such as A54 [108] or AM237 [109] were reported, but have not yet been tested in primary cells or even CFs.

Taken together, our results reveal the dispensability of TRPC proteins for the acute AngII-induced Ca^2+^ signaling in isolated adult murine CFs. Furthermore, they prove the importance of cell density for intracellular Ca^2+^ homeostasis and transdifferentiation. CFs cultured at low density show markedly increased expression of the myofibroblast-marker α-SMA. Our work suggests that other Ca^2+^ conducting channels such as TRPV4, TRPM2, TRPM7 or members of the ORAI family of Ca^2+^ channels may mediate Ca^2+^ entry following activation of AngII-induced signaling cascades in CFs. Finally, by using the CRAC inhibitor GSK7975A we could show an important contribution of CRAC channels to the AngII-evoked Ca^2+^ signaling in cardiac fibroblasts. 

## Figures and Tables

**Figure 1 cells-09-00322-f001:**
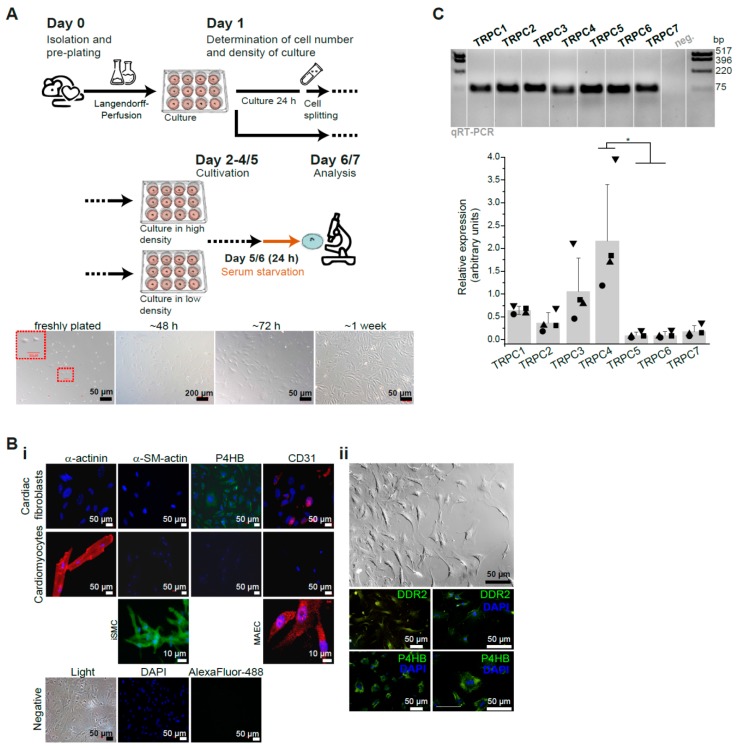
Characterization of primary isolated murine cardiac fibroblasts. (**A**) Isolated primary CFs from adult mice were cultured at high (>300 cells/mm^2^) and low density as depicted. Phase contrast images showing the typical morphology and confluence of cultured fibroblasts at several time points of the culture are included. (**B**) Immunocytochemical characterization of primary cultures of CFs using antibodies against α-actinin, α-smooth muscle (SM) actin, P4HB and CD31 (**Bi**) or additionally DDR2 (**Bii**) is shown. (**C**) Expression analysis by qPCR from *Trpc* channels in four independent (n = 4 hearts) primary cultures of CFs. bp: base pair. * *p* < 0.05 depicted are according to Dunn’s post-hoc comparisons after Kruskal-Wallis.

**Figure 2 cells-09-00322-f002:**
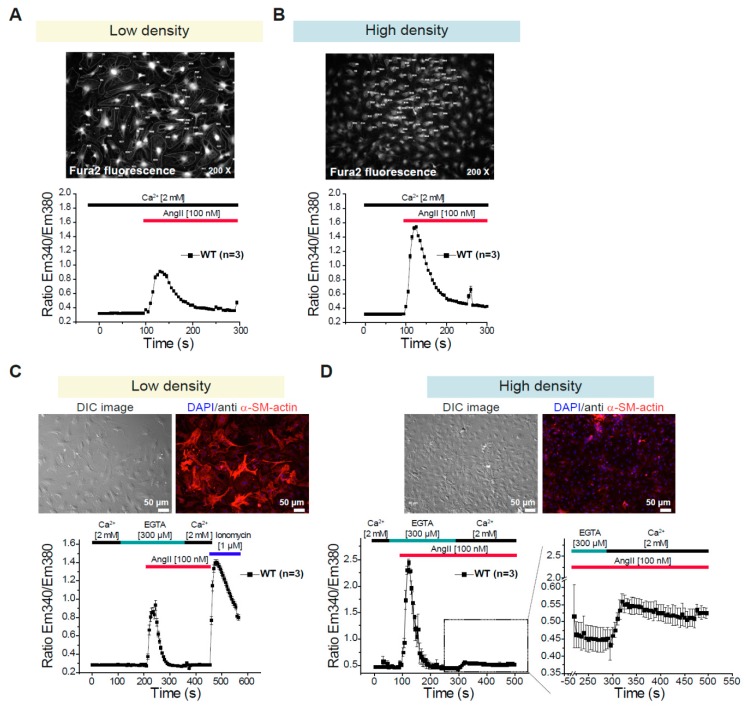
Changes in intracellular Ca^2+^ concentration in cardiac fibroblasts depending on the density of the culture. (**A**,**B**) AngII-induced Ca^2+^ transients in primary CFs in the presence of 2 mM extracellular Ca^2+^. Cells were analyzed 5 days after isolation and cultivation at low (**A**) or high density (**B**). Fluorescence images of Fura-2 loaded cells are included. (**C**,**D**) AngII-induced Ca^2+^ release from internal stores and subsequent AngII-induced Ca^2+^ entry were analyzed in fibroblasts maintained at low (**C**) or high density (**D**) conditions. Additionally, the expression of α-SM-actin was analyzed on cells cultured on each cell density (upper panels). n = number of independent preparations (hearts).

**Figure 3 cells-09-00322-f003:**
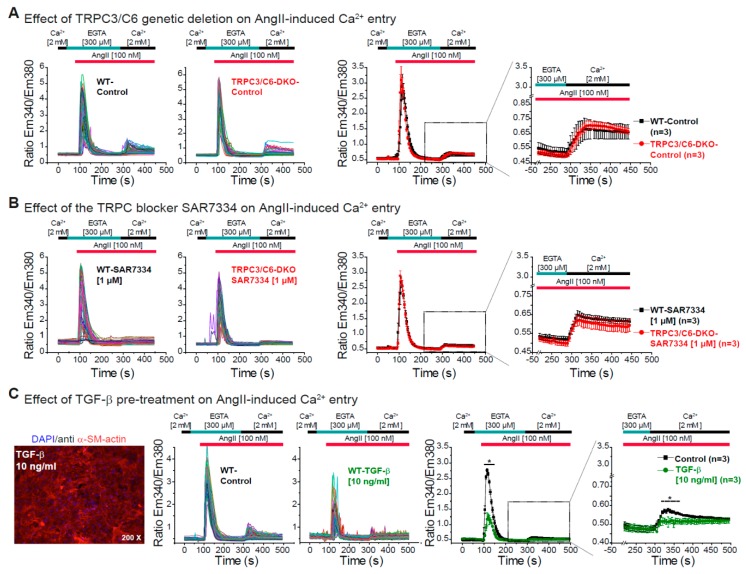
AngII-induced Ca^2+^ release and Ca^2+^ entry in the absence of TRPC3/C6 or after TGF-β pre-treatment. (**A**) AngII-induced Ca^2+^ release and Ca^2+^ entry in primary CFs from WT (black) and TRPC3/C6-DKO (red) mice. Ca^2+^ release was measured in the absence of extracellular Ca^2+^ (300 µM EGTA) and Ca^2+^ entry was monitored in the presence of 2 mM extracellular Ca^2+^. Left panels: Original traces and right panels: Mean values from three independent preparations (hearts). (**B**) Measurements performed as in (A) but in cells pre-incubated (10 min) with the TRPC3/C6/C7 antagonist SAR7334 (1 µM). (**C**) AngII-induced Ca^2+^ release and Ca^2+^ entry in primary CFs from WT mice cultivated in the presence of 10 ng/mL TGF-β (green) or under control conditions (black). Left panel: α-actin smooth muscle staining after TGF-β treatment, middle panels: Original traces from Ca^2+^ measurements and right panels: Mean values from 3 independent preparations. n = number of independent preparations (hearts). All cells were analyzed 5 days after isolation and were cultured at high density. * *p* < 0.05 according to the unpaired Student’s *t*-test.

**Figure 4 cells-09-00322-f004:**
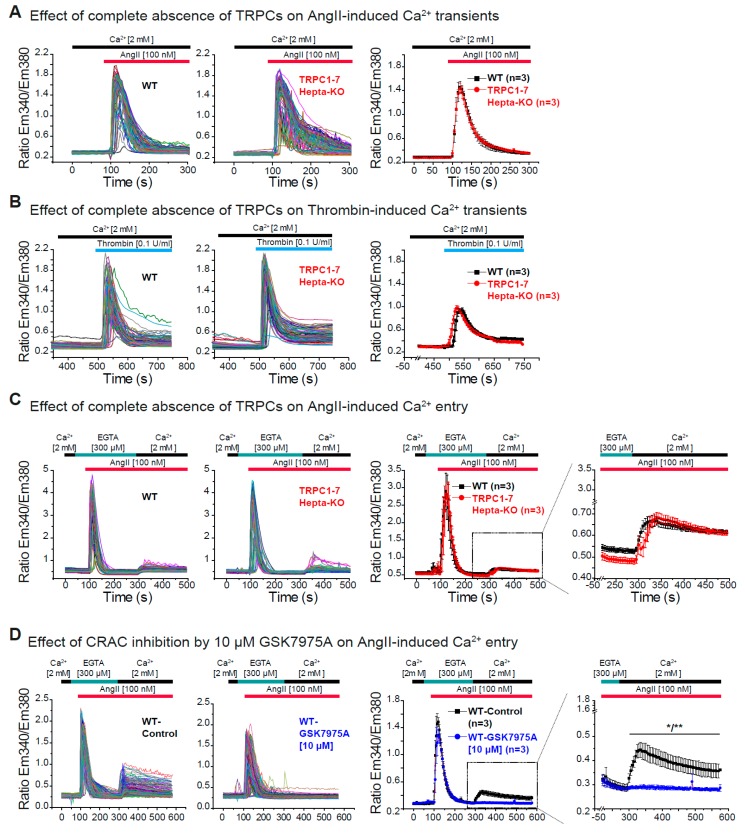
AngII-induced Ca^2+^ release and Ca^2+^ entry in cardiac fibroblasts in the absence of all seven TRPC proteins. (**A**) AngII- and (**B**) thrombin-induced Ca^2+^ transients in primary CFs from WT (black) and TRPC-hepta (*Trpc1/2/3/4/5/6/7*^–/–^) KO (red) mice. Ca^2+^ transients were measured in the presence of 2 mM extracellular Ca^2+^. Left panels: Original traces and right panels: Mean values of three independent preparations (hearts). (**C**) AngII-induced Ca^2+^ release and Ca^2+^ entry in primary CFs from WT (black) and TRPC-hepta KO (red) mice. Ca^2+^ release was measured in the absence of extracellular Ca^2+^ (300 µM EGTA) and Ca^2+^ entry was monitored in the in the presence of 2 mM extracellular Ca^2+^. Left panels: Original traces and right panels: Mean values of three independent preparations. (**D**) The effect of the CRAC blocker (10 µM) GSK7975A on the AngII-induced Ca^2+^ release was analyzed in CFs from WT mice like in (**C**). Left panels: Original traces and right panels: Mean values of three independent preparations. n = number of independent preparations (hearts). All cells were analyzed 5 days after isolation and were cultured at high density. * *p* < 0.05 and ** *p* < 0.01 according to the unpaired Student’s *t*-test.

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
