# Peer review of "Angiotensin-II-Evoked Ca2+ Entry in Murine Cardiac Fibroblasts Does Not Depend on TRPC Channels"

_cells, 2020, doi:10.3390/cells9020322_

Round 1
Reviewer 1 Report
This paper seeks to establish whether canonical TRP (TRPC) ion channels are important for angiotensin II-induced calcium entry in mouse cardiac fibroblasts. The paper is generally well written and clear, the methods are well described and quality of the data is good. The study reports negative results with none of the experiments revealing a role for any of the TRPC channels. The Discussion is overly long in my opinion and strays from a relatively straightforward outcome.
Broad comments:
None of the experiments show any effect of knocking out or inhibiting TRPC channels. Inclusion of selective agonists of TRPC channels in these experiments would have been useful to at least demonstrate coupling of channels to calcium entry and the subsequent efficacy of the KO or pharmacological blockers. As it stands, we don’t learn anything about the role of TRPC channels in cardiac fibroblasts.
The final experiment on cardiac fibroblasts cultured from hepta-KO mice which have had all 7 TRPC channels knocked out is the key experiment. The results of this indicate there is no role for TRPC channels in this setting, and hence no need to perform all the other experiments. One might argue that this should have been performed first, or alternatively a pan-TRPC blocker (or combinations of more specific blockers) should have been used to see if there was any role of TRPC channels before embarking on the rest of the study.
Reviewer 2 Report
In this study, the authors investigated the involvement of TRPC channels in the Angiotensin‐II‐evoked Ca2+ entry in murine cardiac fibroblasts. There are two major findings; 1) the Ca2+ entry functional levels are associated with the culture density and alpha actinic expression level, 2) no member of TRPC subfamily, even the knockout of all 7 members, are responsible for the Ca2+-entry induced by AngII stimulation. The study required elaborate preparation from all different mice lines, and the experimental data are presented in appropriate manner. However, I have a few questions before agreeing to publish the manuscript.
Since the protocol of Ca2+ measurement is typically used to detect the store-operated Ca2+ entry, one can naturally ask whether the Orai/STIM-mediated CRAC channels are involved. However, the authors did not investigate this possibility. I agree that the precise analyses of TRPC members are worth investigating. Nevertheless, general readers would request to check the involvement of Orai/STIM at least. I strongly suggest to test pharmacological inhibitors of Orai/STIM channels. At least, the effects of low concentration of Gd3+ could be checked.2. Although the authors claim the putative importance of the late Ca2+ influx after the AngII stimulation, the amplitude of sustained Ca2+ increase was very small when compared with the peak increase mediated by stored Ca2+ release. Some discussion is requested to this question.
3. Related with the above question, I wonder whether an immediate desensitization (receptor internalization?) of AngII receptor might underlie the relatively small Ca2+ influx. Did the authors check the possibility? In other words, was it possible to evoke second Ca2+ increase after the washout of AngII for the first stimulation. Also, how much time was required for the washout interval to evoke the second response?
Round 2
Reviewer 1 Report
The authors have addressed most of my concerns adequately and their responses regarding the importance of the work in the context of the literature were clear (clearer than in the paper itself). Addition of the data with the CRAC channel blocker is welcomed as this suggests an alternative mechanism for Ang II-induced Ca entry rather than TRPC channels. The new graphical abstract is a succinct summary of the findings. However, as it stands, the manuscript is now lacking important introductory information on CRAC channels (including definition of CRAC) and Orai etc. A few sentences on this should be included in the introduction to fits alongside the ROCE and SOCE which is well described.
Line 539 - should be alpha-SMA not alpha-SMC
Reviewer 2 Report
The authors have revised the manuscript, and added the experiment investigating the role of Orai/STIM (CRAC) channel in the sustained Ca2+ influx induced by AngII. The results are consistent with the conclusion and I generally agree to publish the manuscript in the present state.
Author Response
Thank you to the reviewer for his initial comments and we are happy that we were able to address his questions.